# The Effect of the Digital Economy on the Employment Structure in China

Yantong Zhao [1,2]  and Rusmawati Said [1,*]

1 School of Business and Economics, University Putra Malaysia, Serdang 43400, Selangor, Malaysia; gs60065@student.upm.edu.my
2 Confucian School of Business, Jining University, Qufu 273155, China
* Correspondence: rusmawati@upm.edu.my

**Abstract:** The digital economy's influence on society and the traditional economy is deepening, owing to the wide application of digital technology in production and life. The question of how the digital economy affects the employment structure has become a hot issue to discuss. To explore the impact of the digital economy on the labour structure, this paper selected China's thirty-one provincial panel data between 2013 and 2020 and utilized the static panel model. On the whole, the proportion of employment in the secondary sector to the total employment rate is declining with the development of the digital economy. The proportion of employment in the tertiary sector to the total employment has increased due to enhancement in the degrees of development in the digital economy. From the perspective of different regions, in the eastern and middle part, improvement in the digital economy has reduced the proportion of employment in the primary sectors to the total employment rate while increasing the proportion of employment in the tertiary sector to the total employment rate and optimizing the industrial structure. Employment in the manufacturing and construction industries in the secondary sector is significantly negatively affected by the development of the digital economy. In addition, the mining industry and utility employment (Production and Supply of Gas, Heat, Water, and Electricity) are not significantly affected by the progress of the digital economy.

**Keywords:** digital innovation; digital economy; employment structure; China sectors

## 1. Introduction

We are currently experiencing a new round of technological innovation based on the Internet, and the global economy has entered a new stage of digital economic development (Liu 2022). Governments of various countries have introduced policies to promote the process of national digitalization to support the development of the digital economy. (Barefoot et al. 2018). For instance, different reports have been successively issued by the U.S. government such as "Information Digital Economy 2017" and "Digital Economy 2021" to enrich the connotations associated with the digital economy (Mottaeva et al. 2023). In addition, Canada, Japan, and European Union countries like Germany and Italy have also introduced digital economy strategies to develop their digital economy and serve their national economic development strategies (Panchenko et al. 2020). China remains the leader in developing the global digital economy, with a high concentration of global digital wealth in its commercial platforms (UNCTAD 2021). The digital economy helps improve economic and social life and drives innovation and productivity growth. Consequently, there is a need to continuously guide and emphasize the future impact and development of the digital economy. Parallel to this, this report calls on countries to adopt policy measures to encourage and regulate the development of the digital economy (Dahlman et al. 2016).

The wide application of information and communications technology (ICT), artificial intelligence, big data, and other technological innovations have significantly changed the pattern of the labour market (Said et al. 2008; Vivarelli and Pianta 2000; Vankevich and Kalinouskaya 2021; Zhao et al. 2022). The digital economy's influence on society and

traditional economy is deepening, owing to the wide application of digital technology like big data, cloud computing, and blockchain in production and life; the topics relevant to the digital economy are constantly enriched (Tian et al. 2022). For instance, the association between employment and the digital economy has received extensive attention. Since 2017, UNCTAD's annual "Information Economy Report" (known as the Digital Economy Report since 2019) has studied the association between the digital economy and employment (UNCTAD 2019). The "Information Economy Report 2017" emphasized the direct effect of the digital economy on women's employment (UNCTAD 2017; Hazudin et al. 2021). Consequently, the "Digital Economy Report 2019" appealed to nations to focus more on the potential impact of the digital economy on the employment structure (UNCTAD 2019; Galeeva and Ishtiryakova 2020). Moreover, the effective use of the digital economy to promote employment growth has become a major strategic and economic concern faced by all economies, with the worldwide digital economy entering a phase of rapid development (Autor 2015; Yang 2021).

The digital economy is developing rapidly in China based on the "China Digital Economy Development Report (2022)" issued by the China Academy of Information and Communications Technology (CAICT) (CAICT 2022). The digital economy's added value has recorded an increase from 2.5 trillion Yuan in 2005 to 45.5 trillion Yuan in 2021. As a result, the added value share of the digital economy in GDP was projected to increase from 14% in 2005 to 39.8% in 2021. The development of the digital economy has also accelerated the process of industrial digitization; therefore, its penetration factors in China's agriculture, manufacturing, and service industries were projected to be 8.2%, 19.5%, and 37.8% in 2021, respectively (CAICT 2022). A large number of new jobs and occupations have been created, such as online shopping guides and commentators, online appointment logistics and delivery personnel, and artificial intelligence trainers due to the digital transformation of jobs (Jiang and Murmann 2022).

Since China's population ageing process continues to accelerate, the growth rate of the labour force between 2013 and 2020 declined on a yearly basis after exceeding 0.9% in 2015. Furthermore, the percentage of the population aged 65 and above rose from 9.7 in 2013 to 13.5 in 2020. (Hsu et al. 2018). Meanwhile, accelerating the transition from the demographic dividend to the talent dividend and promoting high-quality economic development have become top priorities. The development of the digital economy enhances the competitiveness of the labour force through the digital application of education (Weninger 2017) and employment skill training programs (Spante et al. 2018), forming a sense of "digital human capital" (Bach et al. 2013). The Fourth Plenary Session of the Nineteenth Central Committee of the Communist Party of China reported that data, as a new production factor other than traditional factors, should be incorporated into the market resource allocation system and the factor reward evaluation mechanism (Guan et al. 2022). Current high-quality economic development has reached a new historical intersection (Zhou et al. 2020). The modern information technology revolution has spawned the rise of many new industrial sectors. Its penetration and integration into the traditional real economy has also led the economy to a new historical position. Therefore, it is of practical significance to investigate the effect of the digital economy's employment structure.

Existing studies have not directly studied the digital economy's effect on the industrial structure of China's labour employment. Based on this, this paper explores how the digital economy impacts the industrial distribution labour force structure in China and provides direct evidence for this theory through empirical testing. Exploring the influence of the digital economy on the industrial structure of labour employment provides a reference point for the future optimization and adjustment of the policy for national employment stabilization.

## 2. Literature Review

The digital economy is defined in "Defining and Measuring the Digital Economy", published by the U.S. Bureau of Economic Analysis in 2018. This explanation defines

the scope of the digital economy from the perspective of the Internet and related ICT industries, including the digital infrastructure required for the existence and operation of computer networks, digital transactions using e-commerce, and content created and accessed by users of the digital economy—i.e., digital media—and on this basis, comprises a detailed list of digital economy products (Barefoot et al. 2018). Digital infrastructure includes telecommunication equipment and services, computer hardware, IoTs, software, buildings, and the services required to support digital infrastructure functions. E-commerce includes B2B, B2C, and P2P. Digital media includes free digital media, direct-sale digital media, and Big Data. From the context of the supply, the BEA expands the meaning of the digital economy based on ICT manufacturing and services, including e-commerce, digital media, and IoTs. Additionally, it is more comprehensive than previous definitions and can better reflect the connotations had with the digital economy. The definition includes the ICT industry and other components with extensive characteristics, lists the product details of the digital economy, and resolves the shortcomings of the blurred boundaries of previous research.

Numerous studies have explored the relationship between digital innovation and employment (Autor 2015; Arntz et al. 2016; Frey and Osborne 2017; Giovannetti and Piga 2023; Damioli et al. 2023). Arntz et al. (2016) indicated the effect of automation on employment in the OECD economies by considering work task heterogeneity based on the task-based method of identifying work tasks. The findings showed that around 10 percent of jobs on average can be replaced by automation in OECD countries. Frey and Osborne (2017) analysed the data for 702 detailed occupations and concluded that about 47% of the US employment is at risk of automation. Nevertheless, Autor (2015) claims that automation complements labour, increasing the demand for labour.

The Petty–Clark Theorem demonstrated that with economic development and improvement of per capita income level, the labour force is transferred from the primary sector to the secondary sector and then to the tertiary sector (Wang et al. 2021; Haris and Said 2012; Haris et al. 2017). Due to their different characteristics, the three sectors have different requirements for technology (Muhammad et al. 2022). As a result, the benefits they can derive from technological progress will vary. A number of authors have considered the effects of the digital economy on the labour structure (Brynjolfsson and McAfee 2014; Sovbetov 2018; Akaev et al. 2020; Petrova et al. 2020; Gardberg et al. 2020; Lu et al. 2023; Jetha et al. 2023). Akaev et al. (2020) suggested that the digital economy drives the tertiarization process. Lu et al. (2023) and Jetha et al. (2023) have studied the impact of the digital economy on gender structure and employment of disabled people. Brynjolfsson and McAfee (2014) believed that the digital economy promoted labour productivity and increased economic growth in America. However, it has reduced the need for traditional labourers, reducing work opportunities (Brynjolfsson and McAfee 2014). Stevenson (2008) studied how the digital economy affects job-searching behaviours. The results show that the Internet makes it far more likely that job seekers will be able to contact employers directly, which is the major reason why the Internet improves job search efficiency. Brynjolfsson and McAfee (2012) pointed out that digital economy application creates a new approach to finding jobs and decreases the cost of finding jobs. Kuhn and Mansour (2014) explored the digital economy's effect on the labour market's matching efficiency by utilizing survey data from the US and pointed out that unemployed individuals who used online job search platforms were reemployed 25% faster than the individuals who did not utilize online platforms for their job search. This confirms that both the employment rate and matching efficiency of the labour market have greatly improved, owing to the digital economy.

In terms of the impact of the development of the digital economy on employment, some of the literature evaluates a certain part of the digital economy such as the impact of the Internet and broadband coverage on the employment structure. The broadband lines used by every 100 individuals in each state were taken as the main explanatory variable by Crandall et al. (2007) to investigate the effect of the digital economy on employment in the US using the least square method. The empirical findings showed that there was

an uplift in employment in different sectors including healthcare, education, and finance industries due to the adoption of broadband. Kolko (2012) used interstate data in the US from 1992 to 2006 and confirmed a direct relationship between broadband coverage and employment growth, especially in industries that heavily rely on technology. Czernich (2014) supposed that the Internet affected employment in two major manners: first, creating more employment opportunities and promoting economic growth; second, stimulating job-matching efficiency. The correlation between the unemployment rate and broadband adoption was analysed in Germany through an empirical study. The results show that broadband coverage was negatively correlated with the unemployment rate. Ivus and Boland (2015) performed a research study on broadband coverage data and employment in Canada for a time period ranging from 1997 to 2011. They concluded that the widespread adoption of broadband promoted rural employment and a wage increase in the service industry, which helped reduce the employment gap between rural and urban areas.

## 3. Methodology

### 3.1. Data and Variables

This paper conducts an empirical analysis based on panel data of thirty-one Chinese provinces (excluding Taiwan, Hong Kong, and Macao) from 2013 to 2020. The research data is derived from the China Statistical Yearbook, the CSMAR database, and the China Labour Statistics Yearbook over the years.

Employment structure (labour). This study takes into account the proportion of employees in three different sectors to the total employment rate in order to measure the labour force's employment structure.

Digital economy development level (digital). Consistent with the measurements of Liu et al. (2020), the digital economy was divided into three major aspects: digital transaction development, information technology development, and Internet development (refer to Table 1 for specific indicators). Fourteen measurement indexes were selected in this paper. The basic data of the measurement indicators were all derived from the China Statistical Yearbook (2013–2020) (National Bureau of Statistics of China 2021). Firstly, the raw data of the variables above were standardized; secondly, the entropy method was used to find out the weight of each indicator in the evaluation system; finally, the comprehensive index of each region was calculated and synthesized to report the digital economy's development level.

**Table 1.** Indicator system of the digital economy.

| First Dimension | Secondary Dimension | Specific Indicators |
|---|---|---|
| Information technology development | Information foundation | Optical cable density |
| | | Mobile phone base station density |
| | | Proportion of informatization practitioners |
| | Informatization influence | Total telecom business |
| | | Software business income |
| Internet development | Fixed internet foundation | Internet access port density |
| | Mobile internet foundation | Mobile Internet penetration rate |
| | Fixed internet impact | Proportion of broadband Internet users |
| | Mobile internet impact | The proportion of mobile Internet users |
| Digital transaction development | Digital trading fundamentals | Number of websites per 100 companies |
| | | Enterprise computer usage |
| | | The proportion of e-commerce enterprises |
| | The impact of digital transactions | E-commerce sales |
| | | Online retail sales |

The density of optical cables, the density of mobile base stations, and the ratio of employees in the information transmission, software, and information technology service industries to the total employment rate in each province were used to measure the investment in informatization. The higher the value of the informatization development index, the higher the informatization level in the province, which was more suitable for developing the digital economy.

As for the measurement of the development level of the Internet, since the core of informatization development is the Internet, this paper mainly measures the development level of the Internet from the perspective of the number of users and the penetration rate, which is more in line with the role of the Internet as a digital economic platform in the context of the digital economy. The higher the Internet development index, the better the construction of the Internet platform in the province, the more Internet user groups, and the more vigorously the digital economy could develop in the province.

For the measurement of the development level of digital transactions, this paper is also divided into two parts: the basis of digital transactions and the impact of digital transactions. On the one hand, digital transactions are inseparable from the use of portals and computers established by enterprises, so we used the ratio of the number of enterprise websites and computers used by enterprises to the number of enterprises in the province as a measurement. On the other hand, the ratio of enterprise activities can measure the degree of emphasis and investment of enterprises in digital transactions to a certain extent, so this paper chose to measure the proportion of e-commerce enterprises in this province. At the same time, this paper selected the province's e-commerce and online retail sales to measure the impact of digital transactions. The larger the value, the larger the scale of digital transactions in the province and the higher the level of digital economy development.

The selected control variables are as follows: the industrial structure upgrade (structure), scaled by the proportion of the tertiary industry's output value to the total output value (Liu et al. 2022); the degree of foreign trade (lntrade), measured by the log of the total import and export volume (Ngouhouo and Nchofoung 2021); the level of urbanization (urban), which is indicated by the ratio of the urban to the total population (Almulhim and Cobbinah 2023); and the economic level (lnGDPpc), proxied by the logarithm of GDP per capita. Table 2 presents the summary of the variable descriptions and data sources for the effect of the digital economy on the employment structure.

**Table 2.** Summary of variable descriptions and data sources.

| Symbol | Variables | Proxy | Data Source |
| --- | --- | --- | --- |
| Primary | Primary sector | the proportion of employees in primary to total employment | China Labour Statistical Yearbook |
| Secondary | Secondary sector | the proportion of employees in secondary to total employment | China Labour Statistical Yearbook |
| Tertiary | Tertiary sector | the proportion of employees in tertiary to total employment | China Labour Statistical Yearbook |
| Digital | Digital economy | Digital economy index | China Statistical Yearbook |
| Structure | Industrial structure | the proportion of the tertiary industry's output value to the total output value. | China Statistical Yearbook |
| Urban | Urbanization | the ratio of the urban to the total population | China Statistical Yearbook |
| lnTrade | the log of trade | the log of the total import and export volume | China Trade and External Economic Statistical Yearbook |
| lnGDPpc | the log of GDP per capita | the log of GDP per capita | China Statistical Yearbook |

### 3.2. Estimation Model

The influence of the digital economy on the employment structure was examined with the econometric model below:

$$labour_{it} = \beta_0 + \beta_1 digital_{it} + \sum_j \beta_j x_{ijt} + \mu_i + \varepsilon_{it}$$

$labour_{it}$ denotes the proportion of employees in different sectors to the total number of employments in the region $i$, $digital_{it}$ represents the development level of the regional digital economy, $x_{ijt}$ stands for the control variables, and $\varepsilon_{it}$ connotes the error term. The unobserved random variable $u_i$ is the intercept term representing individual heterogeneity, namely individual effects.

## 4. Results and Discussions

### 4.1. Benchmark Regression Analysis

Table 3 shows the statistical description of variables, including the names of variables, number of variables, mean value, standard deviation, minimum value, and maximum value.

**Table 3.** Descriptive statistics of variables.

| Variable | Observations | Mean | Std. Dev | Min | Max |
|---|---|---|---|---|---|
| Primary | N = 248 n = 31 T = 8 | 0.32 | 0.14 | 0.02 | 0.63 |
| Secondary | N = 248 n = 31 T = 8 | 0.25 | 0.09 | 0.11 | 0.50 |
| Tertiary | N = 248 n = 31 T = 8 | 0.43 | 0.11 | 0.23 | 0.83 |
| Digital | N = 248 n = 31 T = 8 | 0.22 | 0.12 | 0.07 | 0.77 |
| Structure | N = 248 n = 31 T = 8 | 0.48 | 0.09 | 0.32 | 0.84 |
| Urban | N = 248 n = 31 T = 8 | 0.59 | 0.13 | 0.24 | 0.94 |
| lnTrade | N = 248 n = 31 T = 8 | 17.54 | 1.73 | 13.02 | 20.97 |
| lnGDPpc | N = 248 n = 31 T = 8 | 10.86 | 0.41 | 9.99 | 12.01 |

Table 4 shows a pairwise correlation matrix for the key variables involved in the analysis. There is a positive correlation between the digital economy and proportion of employment in the secondary and tertiary sectors. However, the proportion of employees in the primary sector indicates a negative correlation with the digital economy.

**Table 4.** Correlation matrix for key variables.

|  | **Primary** | **Digital** | **Structure** | **Urban** | **lnTrade** | **lnGDPpc** |
|---|---|---|---|---|---|---|
| Primary | 1.000 | | | | | |
| Digital | −0.670 *** | 1.000 | | | | |
| Structure | −0.567 *** | 0.666 *** | 1.000 | | | |
| Urban | −0.829 *** | 0.584 *** | 0.617 *** | 1.000 | | |
| lnTrade | −0.643 *** | 0.558 *** | 0.228 *** | 0.603 *** | 1.000 | |
| lnGDPpc | −0.864 *** | 0.805 *** | 0.653 *** | 0.840 *** | 0.649 *** | 1.000 |
|  | **Secondary** | **Digital** | **Structure** | **Urban** | **lnTrade** | **lnGDPpc** |
| Secondary | 1.000 | | | | | |
| Digital | 0.352 *** | 1.000 | | | | |
| Structure | −0.076 | 0.666 *** | 1.000 | | | |
| Urban | 0.369 *** | 0.584 *** | 0.617 *** | 1.000 | | |
| lnTrade | 0.695 *** | 0.558 *** | 0.228 *** | 0.603 *** | 1.000 | |
| lnGDPpc | 0.445 *** | 0.805 *** | 0.653 *** | 0.840 *** | 0.649 *** | 1.000 |

**Table 4.** *Cont.*

|  | Tertiary | Digital | Structure | Urban | lnTrade | lnGDPpc |
|---|---|---|---|---|---|---|
| Tertiary | 1.000 |  |  |  |  |  |
| Digital | 0.569 *** | 1.000 |  |  |  |  |
| Structure | 0.807 *** | 0.666 *** | 1.000 |  |  |  |
| Urban | 0.762 *** | 0.584 *** | 0.617 *** | 1.000 |  |  |
| lnTrade | 0.236 *** | 0.558 *** | 0.228 *** | 0.603 *** | 1.000 |  |
| lnGDPpc | 0.741 *** | 0.805 *** | 0.653 *** | 0.840 *** | 0.649 *** | 1.000 |

Note: One star ('*'), Two stars ('**'), Three stars ('***') denote that the corresponding variable is significant at 10%, 5%, 1% level, respectively.

Table 5 reports the results of baseline estimations. Stata 15 was used to perform the Hausman test, which produced a *p*-value less than 0.01, indicating that the original hypothesis of a random effects model could be rejected, allowing the Fixed Effects (FE) model to be used. The clustering robust standard error was used to eliminate the heteroscedasticity and autocorrelation on the model. It could be found that, on the whole, the employment proportion in the secondary sector to the total employment rate declined with the development of the digital economy, and the estimated coefficient displayed statistical significance at a level of 1%. Since the secondary sector was characterized by a high portion of fixed assets, high technology intensity and the digital transformation of skilled labour was complex. Therefore, there was a more significantly negative influence of the digital economy's development on the labour force in the secondary sector. Secondly, the proportion of employment in the tertiary sector to the total employment increased due to enhancement in the digital economy's development degrees, with an estimated coefficient of 1% in significance. This means that advancements in the digital economy promoted employment in the tertiary sector since the digital economy had created a large number of new jobs and occupations in the tertiary sector. Similarly, the technology-intensive degree and the proportion of fixed assets were relatively low. Hence, the difficulty of implementing a digital transformation of the labour force in the tertiary sector was relatively small. Therefore, the proportion of employment in the tertiary sector to total employment increased. Thirdly, the improvement of the digital economy did not significantly account for the employment of the primary sector, and the estimated coefficient was negative. Meanwhile, the progress of the digital economy reduced the proportion of employment in the primary sector to the total employment rate as the primary industry's added value was low compared to the secondary and tertiary sectors. Additionally, the use of digital technologies surged due to advancements in the digital economy, and the primary sector's capacity to absorb employment declined.

**Table 5.** Basic regression result.

| VarName | Primary | Secondary | Tertiary |
|---|---|---|---|
| Digital | −0.013 | −0.186 *** | 0.199 *** |
|  | (−0.38) | (−4.12) | (3.22) |
| Structure | 0.159 ** | −0.193 *** | 0.034 |
|  | (2.27) | (−3.30) | (0.38) |
| Urban | −0.481 *** | 0.308 *** | 0.173 * |
|  | (−3.49) | (3.38) | (1.15) |
| lntrade | 0.003 | −0.004 | 0.001 |
|  | (0.55) | (−0.58) | (0.08) |
| lnGDPpc | −0.058 ** | 0.026 | 0.031 |
|  | (−2.03) | (0.78) | (0.84) |

**Table 5.** *Cont.*

| VarName | Primary | Secondary | Tertiary |
|---|---|---|---|
| _cons | 1.106 *** | −0.008 | −0.098 |
| | (4.22) | (−0.03) | (−0.33) |
| N | 248 | 248 | 248 |
| $R^2$ | 0.75 | 0.45 | 0.73 |
| F | 57.23 | 16.19 | 25.32 |

Note: This table reports the impact of the digital economy on the employment structure. The dependent variable is the proportion of employment in three different sectors to the total employment. All other variables are defined in Section 3.1. The FE model was used based on Hausman test. t statistics are reported in parentheses. ***, **, and * indicate significance at the 1%, 5%, and 10% levels, respectively. Standard errors are in parentheses: *** $p < 0.01$, ** $p < 0.05$, * $p < 0.1$.

Two major methods were considered in this paper to further test the robustness of this study's conclusions. Firstly, the measurement method of the explained variable indicators was changed, and the number of employed persons in different industries was used as a proxy for the labour employment structure variable to re-regress. Secondly, the SAR model and the SEM model were used to re-estimate, while considering that the SAR model controlled the spatial correlation between understudy variables and that the SEM model could potentially control the spatial autocorrelation of the model error term. The above test results confirm that the major conclusions of this paper are still valid, which further indicates the robustness of the study outcomes.

### 4.2. Heterogeneity Analysis

This paper divided the thirty-one provinces into three regions: east, middle, and west to identify regional differences in terms of the digital economy's effect on the industrial structure of labour employment. Furthermore, the principle of combining technological and economic development with geographical location was also adopted in this study. The eastern part includes eleven provinces, including Beijing, Tianjin, Hainan, Guangdong, Shandong, Hebei, Fujian, Liaoning, Shanghai, Zhejiang, and Jiangsu. The central part consists of eight provinces: Jiangxi, Shanxi, Henan, Hunan Anhui, Jilin, Hubei, and Heilongjiang. The western area consists of twelve provinces: Inner Mongolia, Yunnan, Tibet, Shaanxi, Gansu, Qinghai, Guangxi, Chongqing, Sichuan, Guizhou, Ningxia, and Xinjiang. Tables 6–8 demonstrate the results of regression analysis.

The estimations for the eastern region are presented in Table 6. The eastern part represents the most economically developed region. It is also the region with the fastest digital economy development. It is evident that in the eastern part, improvement in the digital economy reduced the employment proportion in the primary and secondary sectors while increasing the employment proportion in the tertiary sector and optimising the industrial structure. The estimated coefficients of the primary and the tertiary sector were significant at a 1% level of statistical significance. Overall, the estimated results of the samples from the eastern region maintained strong consistency with the national-level samples.

**Table 6.** Results of regression in the eastern region.

| VarName | Primary | Secondary | Tertiary |
|---|---|---|---|
| Digital | −0.064 ** | −0.055 | 0.119 ** |
| | (−2.13) | (−1.14) | (2.22) |
| Structure | 0.245 *** | −0.241 *** | −0.004 |
| | (5.03) | (−3.03) | (−0.05) |
| Urban | −0.249 *** | 0.344 *** | −0.095 |
| | (−4.24) | (3.61) | (−0.91) |

**Table 6.** *Cont.*

| VarName | Primary | Secondary | Tertiary |
|---|---|---|---|
| lntrade | 0.003 | 0.024 | −0.026 |
| | (0.25) | (1.33) | (−1.34) |
| lnGDPpc | −0.051 ** | −0.066 * | 0.116 *** |
| | (−2.36) | (−1.87) | (3.01) |
| _cons | 0.774 *** | 0.509 | −0.283 |
| | (3.10) | (1.25) | (−0.63) |
| N | 88 | 88 | 88 |
| $R^2$ | 0.74 | 0.42 | 0.55 |
| F | 166.01 | 69.08 | 21.57 |

Note: This table reports the impact of the digital economy on the employment structure in the eastern region. The dependent variable is the proportion of employment in three different sectors to the total employment rate. All other variables are defined in Section 3.1. The FE model was used based on Hausman test. t statistics are reported in parentheses. ***, **, and * indicate significance at the 1%, 5%, and 10% levels, respectively. Standard errors are in parentheses; *** $p < 0.01$, ** $p < 0.05$, * $p < 0.1$.

Table 7 reports the estimated results for the middle region. It is obvious that the employment proportion in the primary sector to the total employment level was reduced with the advancement of the digital economy in the middle region, and the estimated coefficient was significant, at a 1% level. In terms of coefficients, the impact of the digital economy's development on the employment of the primary sector in the central region was greater than the national average. Secondly, the proportion of employment in the secondary sector in the middle region was reduced by improvement in the digital economy, but the estimated coefficient was not significant as the secondary sector in the central region was underdeveloped. The secondary sector was less affected by the digital economy as compared to the eastern section; therefore, the employment proportion in the secondary sector was not significantly affected by the advancement of the digital economy. Thirdly, the improvement in the digital economy increased the proportion of employment in the tertiary sector, and the estimated coefficient was also significant at a 1% level in statistical significance. Moreover, the promotion of the digital economy to the tertiary sector's employment in the middle region was greater than the national average. The effect of digital economic advancement in the central area on primary, secondary and tertiary jobs was the same as at the national level.

**Table 7.** Results of regression in the middle region.

| VarName | Primary | Secondary | Tertiary |
|---|---|---|---|
| Digital | −0.624 *** | −0.040 | 0.522 *** |
| | (−3.22) | (−0.22) | (2.75) |
| Structure | 0.154 * | 0.019 | −0.201 ** |
| | (1.80) | (0.23) | (−2.23) |
| Urban | 0.261 | −0.605 *** | 0.632 *** |
| | (1.14) | (−2.62) | (3.87) |
| lntrade | 0.017 | 0.004 | −0.028 ** |
| | (1.37) | (0.34) | (−2.16) |
| lnGDPpc | −0.009 | 0.093* | −0.080 * |
| | (−0.19) | (1.95) | (−1.82) |
| _cons | 0.036 | −0.478 | 1.386 *** |
| | (0.08) | (−1.07) | (3.12) |
| N | 64 | 64 | 64 |
| $R^2$ | 0.74 | 0.43 | 0.87 |

Note: This table reports the impact of the digital economy on the employment structure in the middle region. The dependent variable is the proportion of employment in three different sectors to the total employment. All other variables are defined in Section 3.1. The RE model was used based on Hausman test. z statistics are reported in parentheses. ***, **, and * indicate significance at the 1%, 5%, and 10% levels, respectively. Standard errors are in parentheses: *** $p < 0.01$, ** $p < 0.05$, * $p < 0.1$.

Table 8 reports the estimated results for the western region sample. It is notable that the regression results in the western region are consistent with the results at the national level. Digital economic improvement reduced the proportion of employment in the primary and secondary sectors and increased the employment proportion in the tertiary sector, but it was not significant. Compared with the developed eastern regions, the western region generally had the following characteristics: lagging emerging industries, a large proportion of traditional industries, advantages in ecological resources and ecological assets, limited government financial resources, and weak market players. Therefore, there was a limited effect of the digital economy on the western region.

**Table 8.** Results of regression in the western region.

| VarName | Primary | Secondary | Tertiary |
|---------|---------|-----------|----------|
| Digital | −0.008 | −0.067 | 0.075 |
|  | (−0.07) | (−0.56) | (0.46) |
| Structure | 0.179 *** | −0.225 *** | 0.045 |
|  | (3.26) | (−3.96) | (0.58) |
| Urban | −0.985 *** | −0.177 | 1.162 *** |
|  | (−6.64) | (−1.16) | (5.53) |
| lntrade | −0.003 | −0.011 ** | 0.014 ** |
|  | (−0.59) | (−2.24) | (2.05) |
| lnGDPpc | 0.016 | 0.103 *** | −0.118 ** |
|  | (0.49) | (3.11) | (−2.60) |
| _cons | 0.740 ** | −0.518 * | 0.778 * |
|  | (2.50) | (−1.70) | (1.86) |
| N | 96 | 96 | 96 |
| $R^2$ | 0.88 | 0.40 | 0.79 |
| F | 208.41 | 91.31 | 41.99 |

Note: This table reports the impact of the digital economy on the employment structure in the western region. The dependent variable is the proportion of employment in three different sectors to the total employment. All other variables are defined in Section 3.1. The FE model was used based on Hausman test. t statistics are reported in parentheses. ***, **, and * indicate significance at the 1%, 5%, and 10% levels, respectively. Standard errors are in parentheses: *** $p < 0.01$, ** $p < 0.05$, * $p < 0.1$.

The proportion of utility (Production and Supply of Gas, Heat, Water, and Electricity), manufacturing, mining, and construction employment in the secondary sector are separately used as explained variables for regression in order to accurately grasp the digital economy's influence on the employment of specific sub-industries in the secondary sector. The estimated results are depicted in Table 9. Employment in the manufacturing and construction industries in the secondary sector is significantly negatively affected by the development of the digital economy. Nevertheless, the mining industry and utility employment were not significantly affected by the progress of the digital economy. This implies that intelligent equipment derived from digital technology produced a significant substitution effect on employment in the manufacturing and construction industries to a specific extent due to the advancement of the digital economy. Even though the mining and utility industries are labour-intensive industries and the digital economy would have a substitute effect on their employment, the existence of employment compensation offsets the negative effects of the digital economy on their employment.

**Table 9.** Regression results of the second sector segmentation industry.

| VarName | Mining | Manufacturing | Utility | Construction |
|---|---|---|---|---|
| Digital | 0.020 * | −0.119 ** | 0.005 | −0.093 ** |
| | (1.88) | (−3.06) | (0.64) | (−2.56) |
| Structure | −0.047 *** | −0.186 *** | 0.011 | −0.174 *** |
| | (−4.16) | (−4.47) | (1.44) | (−4.48) |
| Urban | −0.098 *** | 0.332 *** | −0.003 | −0.146 ** |
| | (−4.47) | (3.99) | (−0.21) | (−2.11) |
| lntrade | −0.001 | −0.006 | −0.001 | 0.005 |
| | (−0.55) | (−0.98) | (−0.99) | (1.22) |
| lnGDPpc | −0.01 | −0.089 *** | −0.005 | 0.061 *** |
| | (−1.61) | (−3.95) | (−1.11) | (3.02) |
| _cons | 0.231 *** | 1.195 *** | 0.092 ** | −0.426 ** |
| | (4.21) | (5.90) | (2.40) | (−2.30) |
| N | 248 | 248 | 248 | 248 |
| $R^2$ | 0.66 | 0.71 | 0.51 | 0.40 |
| F | | 82.49 | 36.58 | |

Note: This table reports the impact of the digital economy on the employment structure. The dependent variable is the proportion of employment in four different industries to the total employment. All other variables are defined in Section 3.1. Mining and Construction used the RE model, while Manufacturing and Utility used the FE model based on the Hausman test. t statistics are reported in parentheses. ***, **, and * indicate significance at the 1%, 5%, and 10% levels, respectively. Standard errors are in parentheses: *** $p < 0.01$, ** $p < 0.05$, * $p < 0.1$.

### 4.3. Discussion

Previous studies have explored the relationships among the financial opening, wages, and employment structure at the sector level (Yang et al. 2015; Chi 2012). However, few scholars have systematically researched the link between digital transformation and the labour structure of China. This study set out to examine the relationship between the digital economy and the change in the sectoral composition of employment in China. These results corroborate the ideas of Frolov and Lavrentyeva (2019), Akaev et al. (2020), and Zhao (2022) who suggested that the digital economy is one driver of the tertiarization process. The development of the digital economy has reduced the employment proportion in the primary sectors while increasing the employment proportion in the tertiary sector and optimising the industrial structure. A limitation of this study is that the research time is relatively short; only eight years of data are available. A further study could assess the long-term effects of the digital economy on the labour structure.

## 5. Conclusions and Policy Recommendations

### 5.1. Conclusions

From the perspective of digital technology upgrading, China has paid increased attention to developing the digital economy, and there has been continuous improvement in its strategic positioning. Similarly, the digital technology behind the digital economy has brought about profound societal changes. Besides this, China's labour market is bound to be severely impacted, and the labour structure is bound to undergo significant changes. Therefore, this study emphasizes the role of China's digital economy in the employment structure of the labour force. The results show that: (1) On the whole, the employment proportion in the secondary sector to the total employment is declining with the development of the digital economy. The proportion of employment in the tertiary sector to the total employment rate has increased due to the enhancement of the degrees of development in the digital economy. (2) From the perspective of different regions, in the eastern and middle part, improvement in the digital economy has reduced the employment proportion in the primary sectors while increasing the employment proportion in the tertiary sector and optimising the industrial structure. The development of the digital economy in the western region has no significant impact on employment. (3) Employment in the manufacturing and construction industries in the secondary sector is significantly negatively affected by the development of the digital economy. Despite this, the mining

industry and utility employment are not significantly affected by the progress of the digital economy.

### 5.2. Policy Recommendations

First, constant improvement in China's digital economy will increasingly replace labour in the primary and secondary sectors, thereby putting forward new requirements for the quantity and quality of work in the tertiary sector. Therefore, government departments should further increase investment in education funds and the proportion of education funds. In particular, it is essential to give preference to vocational education and carry out job-transfer training for labourers in the primary and secondary sectors. Parallelly, it is necessary to improve interaction and communication mechanisms within the government, training institutions, and labour force, and it is necessary to increase the training of modern service industry employment skills according to the actual demand of the labour force using new formats and new technologies. There is also a need to comply with the upgrading and transformation trends of the labour demand structure, which results from the development of the digital economy, and support the labour force of the primary and secondary sectors in the completion of employment transfers by taking advantage of the situation.

Secondly, the digital economy's current development in the western region requires more promotion measures to avoid falling into a vicious circle of overall backwardness. In addition to this, improvements should be made in digital technology's primary research and development capabilities while strengthening essential core technology research. There is also a need to conduct R&D of major core technologies including operating systems, artificial intelligence, and high-end chips, focusing on original innovation and ecological cultivation.

**Author Contributions:** Conceptualization, Y.Z. and R.S.; methodology, Y.Z. and R.S.; validation, Y.Z.; formal analysis, Y.Z. and R.S.; resources, Y.Z. and R.S.; writing—original draft preparation, Y.Z.; writing—review and editing, Y.Z. and R.S. All authors have read and agreed to the published version of the manuscript.

**Funding:** This research received no external funding.

**Informed Consent Statement:** Not applicable.

**Data Availability Statement:** Not applicable.

**Conflicts of Interest:** The authors declare no conflict of interest.

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
