# Peer review of "The Effect of the Digital Economy on the Employment Structure in China"

_economies, doi:10.3390/economies11090227_

Round 1

Reviewer 1 Report

The paper aims at discussing the change in the sectoral composition of employment in China. In more details, authors refer to the digital transformation as the main driver of the tertiarization process, analysing also the regional distribution/dimension of the effect.

 All in all, the paper’s challenge is undeniable and the manuscript is sufficiently well-rooted and presented. However, I have some concerns I am presenting in the following list:

1) Main hypothesis: the tertiarization of the economies is a secular trend, related to innovation, but not necessarily only to digitalization (see Pianta, M. and Vivarelli, M. (eds) (2000) The employment impact of innovation, Routledge, London). For a better contextualization, a proper reference to a more general trend should be included in the Introduction/Literature review sections.

2) Lines 131-132: When referring to Acemoglu and Restrepo work, authors recall the “compensation theory” that includes also the wages reduction mechanism. Indeed, this mechanism is quite debated due to the decent work perspective of the ILO and the minimun wage on-going dicussions. Therefore, I would properly referring to Acemoglu and Restrepo (a pillar), but making a more critical contextualization/discussion of their results.

3) Literature review: the literature review section needs to be extended including recent works that use more recent data (otherwise, referring to the digital economy with data back in the ‘90s makes the narrative less convincing). See, for instance (with different approaches):

-Cattani, L., Dughera, S. & Landini, F. (2023) Interlocking complementarities between job design and labour contracts. Italian Economic Journal. https://doi.org/10.1007/s40797-022-00192-5

- Damioli, G., Van Roy, V., Vertesy, D., Vivarelli, M. (2023) AI technologies and employment: micro evidence from the supply side, Applied Economics Letters. https://doi.org/10.1080/13504851.2021.202412

-Giovannetti, E., Piga, C. (2023) The multifaceted nature of cooperation for innovation, ICT and innovative outcomes: evidence from UK Microdata. Eurasian Business Review. https://doi.org/10.1007/s40821-023-00241-8

-Goel, R.K., Nelson, M.A. (2022) Employment Effects of R&D and Process innovation: evidence from small and medium-sized firms in emerging markets. Eurasian Business Review. https://doi.org/10.1007/s40821-022-00203-6

-Kastelli, I., Dimas, P., Stamopoulos, D., Tsakanikas, A. (2022) Linking Digital Capacity to Innovation Performance: the Mediating Role of Absorptive Capacity. Journal of Knowledge Economy. https://doi.org/10.1007/s13132-022-01092-w

4) The construction of the “digital variable” (see Table 1) should be more extensively discussed, as this is the main hypothesis of the authors. A FN can be enough.

5) The 248 observations reported in tables should be better clarified both in time dimension and number of regions. I expect 248 if the total number of obs (n x T), but this has to be reported.

6) Table 2 could be extended including basic correlation coefficients. In addition, I would replace (everywhere) ‘lnGDP’ with ‘lnGDPpercapita’ to make Christal clear that the macro variable is a pro capite value.

7) The comment in line 206 “The Hausman test suggests the application of either random-effects or fixed-effects models for estimation” is quite unusual. Hausman test has a defined null hypothesis: it can be rejected or not, but it cannot be ambiguous.

8) Paragraph 4.3: I would report this result in a FN, otherwise this section needs to report the cited evidence/results.

 Minor:

-lines 36-37: reports should be cited following the chronological order

-line 58: Please, make explicit the acronym CAICT

Reviewer 2 Report

The paper has some potential to fill the cognitive gap on the link between digitalization of economy and employment, but it suffers from some minor and major problems.

First of all, the title of the paper doesn’t correspond with its aim and content. As stated by the author(s), “this paper explores how the digital economy impacts the industrial distribution labour force structure in China” (p. 2, lines: 86-87), whereas the title suggests that the paper deals with the effects of  digital innovation on employment structure in China. I want to notice that digital economy is not considered to be equivalent to digital innovation.

The literature review  provides a rapid overview of some studies on the impact of digitalization, including the use of ICT, on employment. Unfortunately, it doesn’t. lead to the formulation of any research hypotheses or questions. As for the scope of the literature review, more attention needs to be paid to the functioning of the quaternary sector that is directly associated with digital economy.

It should be noticed that the methodology of research raises a number of concerns. Firstly, the authors should detect heteroskedasticity and/or autocorrelation before testing which specification (fe- or -re- specification) fits data better. Next, the results of these tests should be presented and discussed. What is surprising, there is the statement that “The Hausman test suggests the application of either random-effects or fixed-effects models for estimation.” (p. 5, lines: 206-207), whereas this test can be applied to differentiate between fixed effects model and random effects model.  Secondly, the general modelling framework for analyzing panel data presented in the section 3.1 is not correct (see: Greene, W. (2018) Econometric Analysis. 8th Edition, Pearson Education Limited, London). Incidentally, this section  shall be placed immediately after the section 3.2.

The presentation of the results is also problematic. For instance, from Tables 3-7  it cannot be inferred  what kind of models are estimated. These tables also don’t include all important measures of models quality. Moreover, the results of the robustness tests are not presented.

The interpretation of results is very schematic and not contrasted with other studies.

The paper is readable, but it requires some copy-editing to fix various errors.

Reviewer 3 Report

The topic of the paper is interesting. The article explores the influence of the digital economy on the employment structure in China. However, there is a need for a thorough revision.

Introduction section needs a revision, to adequately introduce the topic:

-        Update the references – for some reports there are new editions available (eg UNCTAD’s Information Economy Report).

-        Also, the figures on China’s labor force could be updated; lines 68-84 describe the structure for 2010 – 2018, and, according to the author(s), the paper is focused on 2013 – 2020.

-        First and second paragraphs need a better structure to help the reader to follow the ideas. Eg – in the first paragraph – the author(s) intention was to emphasize that the UNCTAD report ask for policy measures or just to say that policy measures are important? For the second paragraph – the idea was to give some examples of countries and regions that have policies related to digital economy; why they choose such old examples for USA? Germany, Italy are members of the European Union – the phrase needs a revision. Etc.

-        Lines 76-79 quotation needed if it is about a policy paper (add online resource or provide a reliable source for the information).

The theoretical background shows that the authors have reviewed the literature; however, this section lacks a clear organizational structure and thus an argumentative line.

Methodology section should be developed, particularly Data and Variable Section where important information is missing.

There is a mismatch between the concept used in the title “digital innovation” and the one used in the paper “digital economy”.

I am not qualified to assess the quality of English. Nevertheless, moderate editing of English language is required in my opinion because there are some words/phrases unusual for standard English or difficult to understand: Abstract – “This paper uses the static 11 panel model to adopt China's thirty-one provincial panel data between 2013 and 2020.”; Inroduction – “Several policies have been introduced to promote the process of digitalization due to the full play of the digital economy’s role.”, etc.

The results and discussion are satisfactory. In the conclusion, I would have liked to have a discussion on limits of the research and futures directions of research. 

I believe that the authors should provide a greater number of more recent bibliographical references. Of the total number of references, less than 15% are works published in the last two years. 

Round 2

Reviewer 2 Report

I do appreciate that some of my comments have been addressed by the authors, but I am still not entirely satisfied with replies to the methodological issues. First of all, it seems to me that the authors ignore the specificities of panel data. As I suggested, the authors should test  for panel-level heteroskedasticity and autocorrelation (see: https://www.stata.com/support/faqs/statistics/panel-level-heteroskedasticity-and-autocorrelation/). By the way, I want to notice that the White test is used in time series data. Moreover, the calculations of the pairwise correlation (Table 4) and VIFs are also problematic. In the first case, it is not clear whether the authors calculate a correlation for all the values of two variables in a dataset (pooling panels) or control for correlation properties of the panels. In the second case, the VIFs  should be checked for transformed predictors with individual-specific means subtracted when estimating a fixed effects model. This is not an issue for doing random effects.

In my opinion, the authors should address the issue, which concerns the relationships between dependent variables in the basic regressions (Table 5) and the region specific regressions (Table 6, 7 and 8).  Given that the shares of employees in three different sectors total 100%, the coefficients for a particular variable in three regressions  sum to zero. A such, one coefficient can be expressed by the two others linearly.

Finally, the presentation of the econometric model (the section 3.2) should allow for capturing the difference between fixed and random effects models.

The quality of English language is acceptable.

Round 3

Reviewer 2 Report

The authors have addressed all my concerns.

Minor editing of English language is required.